# FEATURE MAP VARIATIONAL AUTO-ENCODERS

## ABSTRACT

There have been multiple attempts with variational auto-encoders (VAE) to learn powerful global representations of complex data using a combination of latent stochastic variables and an autoregressive model over the dimensions of the data. However, for the most challenging natural image tasks the purely autoregressive model with stochastic variables still outperform the combined stochastic-autoregressive models. In this paper, we present simple additions to the VAE framework that generalize to natural images by embedding spatial information in the stochastic layers. We significantly improve the state-of-the-art results on MNIST, OMNIGLOT, CIFAR10 and ImageNet when the feature map parameterization of the stochastic variables are combined with the autoregressive PixelCNN approach. Interestingly, we also observe close to state-of-the-art results without the autoregressive part. This opens the possibility for high quality image generation with only one forward-pass.

## 1 INTRODUCTION

In representation learning the goal is to learn a posterior latent distribution that explains the observed data well (Bengio et al., 2013). Learning good representations from data can be used for various tasks such as generative modelling and semi-supervised learning (Kingma, 2013; Rezende et al., 2014; Kingma et al., 2014; Rasmus et al., 2015; Maaløe et al., 2016). The decomposition of variational auto-encoders (VAE) (Kingma, 2013; Rezende et al., 2014) provides the potential to disentangle the internal representation of the input data from local to global features through a hierarchy of stochastic latent variables. This makes the VAE an obvious candidate for learning good representations. However, in order to make inference tractable VAEs contain simplifying assumptions. This limits their ability to learn a good posterior latent representation.

In complex data distributions with temporal dependencies (e.g. text, images and audio), the VAE assumption on conditional independence in the input distribution limits the ability to learn local structures. This has a significant impact on its generative performance, and thereby also the learned representations. Additionally, the one-layered VAE model with a $\mathcal{N}(0, I)$ latent prior poses serious constraints on the posterior complexity that the model is able to learn. A deep hierarchy of stochastic latent variables should endow the model with more expressiveness, but the VAE has a tendency to *skip* the learning of the higher representations since they pose a direct cost in its optimization term.

There have been several attempts to eliminate the limitations of the VAE. Some concern formulating a more expressive variational distribution (Burda et al., 2015b; Rezende & Mohamed, 2015; Tran et al., 2016; Maaløe et al., 2016) where other concerns learning a deeper hierarchy of latent variables (Sønderby et al., 2016). These contributions have resulted in better performance, but are still limited when modelling complex data distributions where a conditional independence does not apply. When parameterizing the VAE decoder with recurrent neural networks (Krishnan et al., 2015; Bowman et al., 2015; Fraccaro et al., 2016), the decoding architecture gets too powerful which results in unused latent stochastic variables (Chen et al., 2017).

The limitations of the VAE have spawned interest towards other generative models such as Generative Adversarial Networks (GAN) (Goodfellow et al., 2014) and the autoregressive Pixel-CNN/PixelRNN models (van den Oord et al., 2016b). These methods have proven powerful in learning good generative models, but the lack of stochastic latent variables makes them less suitable for representation learning purposes (Chen et al., 2017). Lately, we have seen several successful attempts to combine VAEs with PixelCNNs (Gulrajani et al., 2016; Chen et al., 2017). This results

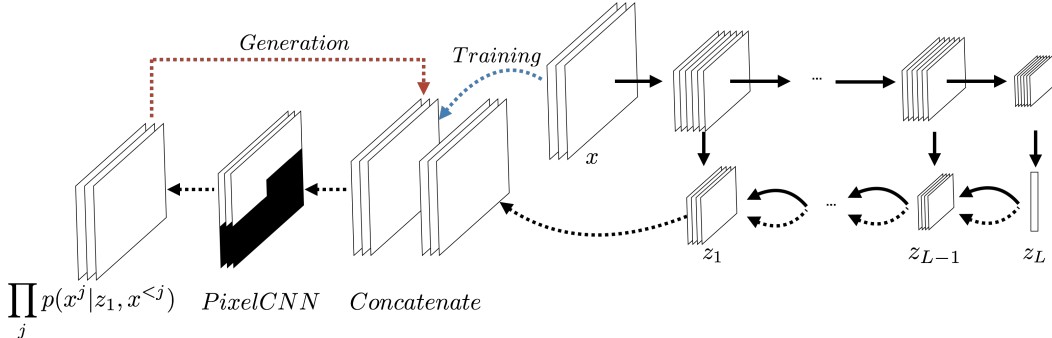

Figure 1: A visualization of FAME where the solid lines denote the variational approximation (inference/encoder/recognition) network and dashed lines denote the generative model (decoder) network for training. When performing reconstructions during training, the input image is concatenated with the output of the generative model (blue) and when generating the model follows a normal autoregressive sampling flow (red) while also using the stochastic latent variables $\mathbf{z} = z_1, ..., z_L$. Both the variational approximation and the generative model follow a top-down hierarchical structure which enables precision weighted stochastic variables in the variational approximation.

in a model where the global structure of the data is learned in the stochastic latent variables of the VAE and the local structure is learned in the PixelCNN. However, despite the additional complexity and potential extra expressiveness, these models do not outperform a simple autoregressive model (van den Oord et al., 2016a; Salimans et al., 2017).

In this paper we present the Feature Map Variational Auto-Encoder (FAME) that combines the top-down variational approximation presented in the Ladder Variational Auto-Encoder (LVAE) (Sønderby et al., 2016) with a spatial (feature map) representation of the stochastic latent variables and an autoregressive decoder. We show that (i) FAME outperforms previously state-of-the-art log-likelihood on MNIST, OMNIGLOT, CIFAR10 and ImageNet, (ii) FAME learns a deep hierarchy of stochastic latent variables without inactivated latent units, (iii) by removing the autoregressive decoder FAME performs close to previous state-of-the-art log-likelihood suggesting that it is possible to get good quality generation with just one forward pass.

## 2 FEATURE MAP VARIATIONAL AUTO-ENCODER

The VAE (Rezende et al., 2014; Kingma, 2013) is a generative model with a hierarchy of stochastic latent variables:

$$p_\theta(x, \mathbf{z}) = p_\theta(x|z_1)p_\theta(z_L) \prod_{i=1}^{L-1} p_\theta(z_i|z_{i+1}) \,, \tag{1}$$

where $\mathbf{z} = z_1, ..., z_L$, $\theta$ denotes the parameters, and $L$ denotes the number of stochastic latent variable layers. The stochastic latent variables are usually modelled as conditionally independent Gaussian distributions with a diagonal covariance:

$$p_\theta(z_i|z_{i+1}) = \mathcal{N}\Big(z_i; \mu_{\theta,i}(z_{i+1}), \mathrm{diag}(\sigma_{\theta,i}^2(z_{i+1}))\Big), \qquad p_\theta(z_L) = \mathcal{N}\Big(z_L; 0, I\Big) \,. \tag{2}$$

Since the posterior $p(\mathbf{z}|x)$ often is intractable we introduce a variational approximation $q_\phi(\mathbf{z}|x)$ with parameters $\phi$. In the original VAE formulation $q_\phi(\mathbf{z}|x)$ is decomposed as a bottom-up inference path through the hierarchy of the stochastic layers:

$$q_\phi(\mathbf{z}|x) = q_\phi(z_1|x) \prod_{i=2}^{L} q_\phi(z_i|z_{i-1}) \,, \tag{3}$$

$$q_\phi(z_1|x) = \mathcal{N}\Big(z_1; \mu_{\phi,1}(x), \mathrm{diag}(\sigma_{\phi,1}^2(x))\Big) \,, \tag{4}$$

$$q_\phi(z_i|z_{i-1}) = \mathcal{N}\Big(z_i; \mu_{\phi,i}(z_{i-1}), \mathrm{diag}(\sigma_{\phi,i}^2(z_{i-1}))\Big) \,. \tag{5}$$

We optimize an *evidence lower-bound* (ELBO) to the log-likelihood $\log p_\theta(x) = \log \int_{\mathbf{z}} p_\theta(x, \mathbf{z}) d\mathbf{z}$. Burda et al. (2015a) introduced the importance weighted bound:

$$\log p(x) \geq \mathbb{E}_{q_\phi(\mathbf{z}^1|x)}, ..., \mathbb{E}_{q_\phi(\mathbf{z}^K|x)} \left[ \log \sum_{k=1}^K \frac{p_\theta(x, \mathbf{z}^k)}{q_\phi(\mathbf{z}^k|x)} \right] \equiv \mathcal{L}_K(\theta, \phi; x) \tag{6}$$

and proved that $\mathcal{L}_K(\theta, \phi; x) \geq \mathcal{L}_L(\theta, \phi; x)$ for $K > L$. For $K = 1$ the bound co-incides with the standard ELBO: $\mathcal{L}(\theta, \phi; x) = \mathcal{L}_1(\theta, \phi; x)$. The hierarchical structure of both the variational approximation and generative model give the VAE the expressiveness to learn different representations of the data throughout its stochastic variables, going from local (e.g. edges in images) to global features (e.g. class specific information). However, we can apply as recursive argument Maaløe et al. (2017) to show that when optimizing with respect to the parameters $\theta$ and $\phi$ the VAE is regularized towards $q_\phi(z_L|z_{L-1}) = p_\theta(z_L) = \mathcal{N}(z_L; 0, I)$. This is evident if we rewrite Equation 6 for $K = 1$:

$$\mathcal{L}(\theta, \phi; x) = \mathbb{E}_{q_\phi(z_{1:L-1}|x)} \left[ \frac{p_\theta(x, z_{1:L-1}|z_L)}{q_\phi(z_{1:L-1}|x)} \right] - \mathbb{E}_{q_\phi(z_{1:L-1}|x)} \left[ KL\big(q_\phi(z_L|z_{L-1})||p_\theta(z_L)\big) \right] .$$

$KL\big(q_\phi(z_L|z_{L-1})||p_\theta(z_L)\big) = 0$ is a local maxima and learning a useful representation in $z_L$ can therefore be disregarded throughout the remainder of the training. The same argumentation can be used for all subsequent layers $z_{2:L}$, hence the VAE has a tendency to collapse towards not using the full hierarchy of latent variables. There are different ways to get around this tendency, where the simplest is to down-weight the $KL$-divergence with a temperature term (Bowman et al., 2015; Sønderby et al., 2016). This term is applied during the initial phase of optimization and thereby downscales the regularizing effect. However, this only works for a limited number of hierarchically stacked latent variables (Sønderby et al., 2016).

Formulating a deep hierarchical VAE is not the only cause of inactive latent variables, it also occurs when the parameterization of the decoder gets too powerful (Krishnan et al., 2015; Fraccaro et al., 2016; Chen et al., 2017). This can be caused by using autoregressive models such as $p(x, \mathbf{z}) = \prod_j p(x^j|x^{<j}, \mathbf{z})p(\mathbf{z})$. Chen et al. (2017) circumvent this by introducing the Variational Lossy Auto-Encoder (VLAE) where they define the architecture for the VAE and autoregressive model such that they capture global and local structures. They also utilize the power of more expressive posterior approximations using inverse autoregressive flows (Rezende & Mohamed, 2015; Kingma et al., 2016). In the PixelVAE, Gulrajani et al. (2016) takes a similar approach to defining the generative model but makes a simpler factorizing decomposition in the variational approximation $q_\phi(\mathbf{z}|x) = \prod_i^L q_\phi(z_i|x)$, where the terms have some degree of parameter sharing. This formulation results in a less flexible model.

In Kingma et al. (2016); Gulrajani et al. (2016); Chen et al. (2017) we have seen that VAEs with simple decompositions of the stochastic latent variables and a powerful autoregressive decoder can result in good generative performance and representation learning. However, despite the additional cost of learning a VAE we only see improvement in the log-likelihood over the PixelCNN for small gray-scale image datasets (Salimans et al., 2017). We propose FAME that extends the VAE with a top-down variational approximation similar to the LVAE (Sønderby et al., 2016) combined with spatial stochastic latent layers and an autoregressive decoder, so that we ensure expressive latent stochastic variables learned in a deep hierarchy (cf. Figure 1).

## 2.1 TOP-DOWN VARIATIONAL APPROXIMATION

The LVAE (Sønderby et al., 2016) does not change the generative model but changes the variational distribution to be top-down like the generative model. Furthermore the variational distribution shares parameters with the generative model which can be viewed as a precision-weighted (inverse variance) combination of information from the prior and data distribution. The variational approximation is defined as:

$$q_\phi(\mathbf{z}|x) = q_\phi(z_L|x) \prod_{i=1}^{L-1} q_\phi(z_i|z_{i+1}, x) . \tag{7}$$

The stochastic latent variables are all fully factorized Gaussian distributions and are therefore modelled by $q_\phi(z_i|z_{i+1}, x) = \mathcal{N}(z_i|\mu_i, \text{diag}(\sigma_i^2))$ for layers $i = 1, ..., L$. Instead of letting $q$ and $p$ have

separate parameters (as in the VAE), the LVAE let the mean and variance be defined in terms of a function of $x$ (the bottom-up data part) and the generative model (the top-down prior):

$$\mu_i = \frac{\mu_{\phi,i}\sigma_{\phi,i}^{-2} + \mu_{\theta,i}\sigma_{\theta,i}^{-2}}{\sigma_{\phi,i}^{-2} + \sigma_{\theta,i}^{-2}} \tag{8}$$

$$\sigma_i = \frac{1}{\sigma_{\phi,i}^{-2} + \sigma_{\theta,i}^{-2}} \, , \tag{9}$$

where $\mu_{\phi,i} = \mu_{\phi,i}(x)$ and $\mu_{\theta,i} = \mu_{\theta,i}(z_{i+1})$ and like-wise for the variance functions. This precision weighted parameterization has previously yielded excellent results for densely connected networks (Sønderby et al., 2016).

## 2.2 CONVOLUTIONAL STOCHASTIC LAYERS

We have seen multiple contributions (e.g. Gulrajani et al. (2016)) where VAEs (and similar models) have been parameterized with convolutions in the deterministic layers $h_j^i$, for $j = 1, ..., M$, and $M$ is the number of layers connecting the stochastic latent variables $z_i$. The size of the spatial feature maps decreases towards higher latent representations and transposed convolutions are used in the generative model. In FAME we propose to extend this notion, so that each of the stochastic latent layers $z_i, ..., z_{L-1}$ are also convolutional. This gives the model more expressiveness in the latent layers, since it will keep track of the spatial composition of the data (and thereby learn better representations). The top stochastic layer $z_L$ in FAME is a fully-connected dense layer, which makes it simpler to condition on a non-informative $\mathcal{N}(0, I)$ prior and sample from a learned generative model $p_\theta(x, \mathbf{z})$. For the $i = 1, ..., L - 1$ stochastic latent variables, the architecture is as follows:

$$h_{M,i} = \texttt{CNN}(h_{<M,i})$$
$$\mu_{\phi \vee \theta,i} = \texttt{Linear}(\texttt{CONV}(h_{M,i}))$$
$$\sigma_{\phi \vee \theta,i} = \texttt{Softplus}(\texttt{CONV}(h_{M,i})) \, ,$$

where `CNN` and `CONV` denote a convolutional neural network and convolutional layer respectively. The top-most latent stochastic layer $z_L$ is computed by:

$$h_{M,L} = \texttt{Flatten}(\texttt{CNN}(h_{<M,L}))$$
$$\mu_{\phi \vee \theta,L} = \texttt{Linear}(\texttt{Dense}(h_{M,L}))$$
$$\sigma_{\phi \vee \theta,L} = \texttt{Softplus}(\texttt{Dense}(h_{M,L})) \, .$$

This new feature map parameterization of the stochastic layers should be viewed as a step towards a better variational model where the test ELBO and the amount of activated stochastic units are direct meaures hereof.

## 2.3 AUTOREGRESSIVE DECODING

From van den Oord et al. (2016b;a); Salimans et al. (2017) we have seen that the PixelCNN architecture is very powerful in modelling a conditional distribution between pixels. In FAME we introduce a PixelCNN in the input dimension of the generative model $p_\theta(x|\mathbf{z})$ (cf. Figure 1). During training we concatenate the input with the reconstruction data in the channel dimension and propagate it through the PixelCNN, similarly to what is done in Gulrajani et al. (2016). When generating samples we fix a sample from the stochastic latent variables and generate the image pixel by pixel autoregressively.

## 3 EXPERIMENTS

We test FAME on images from which we can compare with a wide range of generative models. First we evaluate on gray-scaled image datasets: statically and dynamically binarized MNIST (LeCun et al., 1998) and OMNIGLOT (Lake et al., 2013). The OMNIGLOT dataset is of particular interest due to the large variance amongst samples. Secondly we evaluate our models on natural image datasets: CIFAR10 (Krizhevsky, 2009) and 32x32 ImageNet[1] (van den Oord et al., 2016b). When

---

[1]http://image-net.org/small/download.php

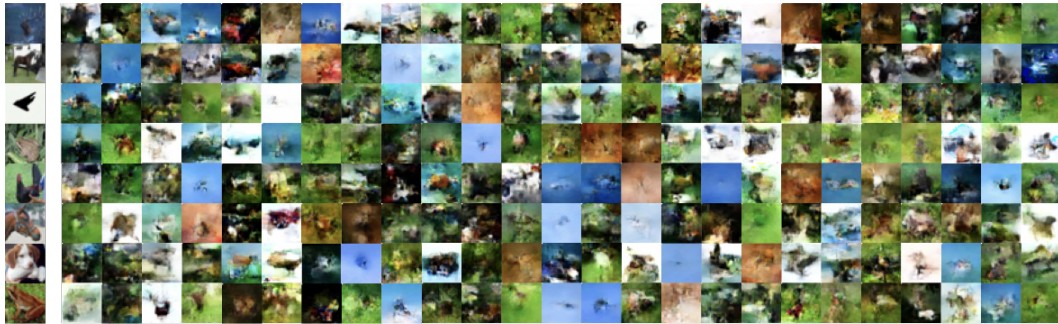

Figure 2: 10 randomly picked CIFAR10 images (left) and 200 random samples drawn from a $\mathcal{N}(0, I)$ distribution and propagated through the generative model (right).

| | NLL | | NLL |
|---|---|---|---|
| IWAE (BURDA ET AL., 2015A) | 82.90 | | |
| LVAE (SØNDERBY ET AL., 2016) | 81.74 | DRAW (GREGOR ET AL., 2015) | 80.97 |
| CAGEM (MAALØE ET AL., 2017) | 81.60 | DVAE (ROLFE, 2017) | 81.01 |
| DVAE (ROLFE, 2017) | 80.04 | IAF VAE (KINGMA ET AL., 2016) | 79.88 |
| VGP (TRAN ET AL., 2016) | 79.88 | PIXELRNN (VAN DEN OORD ET AL., 2016B) | 79.20 |
| IAF VAE KINGMA ET AL. (2016) | 79.10 | VLAE (CHEN ET AL., 2017) | 79.03 |
| VLAE CHEN ET AL. (2017) | 78.53 | PIXELVAE (GULRAJANI ET AL., 2016) | **79.02** |
| FAME NO CONCATENATION | 78.73 | FAME | 79.30 |
| FAME | **77.82** | | |

Table 2: Negative $\log$-likelihood performance on dynamically (left) and statically (right) binarized MNIST in nats. For the dynamically binarized MNIST results show the results for the FAME No Concatenation that has no dependency on the input image. The evidence lower-bound is computed with 5000 importance weighted samples $\mathcal{L}_{5000}(\theta, \phi; x)$.

modelling the gray-scaled images we assume a Bernoulli $\mathcal{B}$ distribution using a Sigmoid activation function as the output and for the natural images we assume a Categorical distribution $\pi$ by applying the 256-way Softmax approach introduced in van den Oord et al. (2016b). We evaluate the gray-scaled images with $\mathcal{L}_{5000}$ (cf. Equation 6) and due to runtime and space complexity we evaluate the natural images with $\mathcal{L}_{1000}$.

We use a hierarchy of 5 stochastic latent variables. In case of gray-scaled images the stochastic latent layers are dense with sizes 64, 32, 16, 8, 4 (equivalent to Sønderby et al. (2016)) and for the natural images they are spatial (cf. Table 1). There was no significant difference when using feature maps (as compared to dense layers) for modelling gray-scaled images. We apply batch-normalization (Ioffe & Szegedy, 2015) and ReLU activation functions as the non-linearity between all hidden layers $h_{i,j}$ and use a simple PixelCNN as in van den Oord et al. (2016b) with 4 residual blocks.

Because of the concatenation in the autoregressive decoder (cf. Figure 1), generation is a cumbersome process that scales linearly with the amount of pixels in the input image. Therefore we have defined a slightly changed parameterization denoted FAME No Concatenation, where the concatenation with the input is omitted. The generation has no dependency on the input data distribution and can therefore be performed in one forward-pass through the generative model.

For optimization we apply the Adam optimizer (Kingma & Ba, 2014) with a constant learning rate of 0.0003. We use 1 importance weighted sample and temperature (Sønderby et al., 2016) scaling from .3 to 1. during the initial 200 epochs for gray-scaled images and .01 to 1. during the first 400 epochs for natural images. All models are trained using the same optimization scheme.

### 3.1 GENERATIVE PERFORMANCE ON GRAY-SCALED IMAGES

The MNIST dataset serves as a good sanity check and has a myriad of previously published generative modelling benchmarks. We experienced much faster convergence rate on FAME compared to training a regular LVAE. On the dynamically binarized MNIST dataset we see a sig-

|  | GRAY-SCALED IMAGES 28X28 | NATURAL IMAGES 32X32 |
|---|---|---|
| $h_{:,1}$ | 1 X CONV F=5X5, K=32, S=2 | 2 X CONV F=3X3, K=96, S=1 |
|  | 1 X CONV F=3X3, K=64, S=1 | 1 X CONV F=3X3, K=96, S=2 |
| $z_1$ | 1 X DENSE D=64 | 1 X CONV F=3X3, K=8, S=1 |
|  | 64 FEATURE VECTOR | 16 X 16 X 8 FEATURE MAPS |
| $h_{:,2}$ | 1 X CONV F=3X3, K=64, S=2 | 2 X CONV F=3X3, K=192, S=1 |
|  | 1 X CONV F=3X3, K=64, S=1 | 1 X CONV F=3X3, K=192, S=2 |
| $z_2$ | 1 X DENSE D=32 | 1 X CONV F=3X3, K=16, S=1 |
|  | 32 FEATURE VECTOR | 8 X 8 X 16 FEATURE MAPS |
| $h_{:,3}$ | 1 X CONV F=3X3, K=64, S=2 | 2 X CONV F=3X3, K=192, S=1 |
|  | 1 X CONV F=3X3, K=64, S=1 | 1 X CONV F=3X3, K=192, S=2 |
| $z_3$ | 1 X DENSE D=16 | 1 X CONV F=3X3, K=16, S=1 |
|  | 16 FEATURE VECTOR | 4 X 4 X 16 FEATURE MAPS |
| $h_{:,4}$ | 1 X CONV F=3X3, K=64, S=2 | 2 X CONV F=3X3, K=192, S=1 |
|  | 1 X CONV F=3X3, K=64, S=1 | 1 X CONV F=3X3, K=192, S=2 |
| $z_4$ | 1 X DENSE D=8 | 1 X CONV F=3X3, K=16, S=1 |
|  | 8 FEATURE VECTOR | 2 X 2 X 16 FEATURE MAPS |
| $h_{:,5}$ | 1 X CONV F=3X3, K=64, S=2 | 2 X CONV F=3X3, K=192, S=1 |
|  | 1 X CONV F=3X3, K=64, S=1 | 1 X CONV F=3X3, K=192, S=2 |
| $z_5$ | 1 X DENSE D=4 | 1 X DENSE D=64 |
|  | 4 FEATURE VECTOR | 64 FEATURE VECTOR |

Table 1: The convolutional layer (Conv), filter size (F), depth (K), stride (S), dense layer (Dense) and dimensionality (D) used in defining FAME for gray-scaled and natural images. The architecture is defined such that we ensure dimensionality reduction throughout the hierarchical stochastic layers. The autoregressive decoder is a PixelCNN (van den Oord et al., 2016b) with a mask *A* convolution F=7x7, K=64, S=1 followed by 4 residual blocks of convolutions with mask *B*, F=3x3, K=64, S=1. Finally there are three non-residual layers of convolutions with mask *B* where the last is the output layer with a Sigmoid activation for gray-scaled images and a 256-way Softmax for natural images.

nificant improvement (cf. Table 2). However, on the statically binarized MNIST, the parameterization and current optimization strategy was unsuccessful in achieving state-of-the-art results (cf. Table 1). In Figure 4a we see random samples drawn from a $\mathcal{N}(0, I)$ distribution and propagated through the decoder parameters $\theta$. We also trained the FAME No Concatenation which performs nearly on par with the previously state-of-the-art VLAE model (Chen et al., 2017) that in comparison utilizes a skip-connection from the input distribution to the generative decoder: $p_{\text{local}}(x|z) = \prod_i p(x_i|z, x_{\text{WindowAround}(i)})$. This proves that a better parameterization of the VAE improves the performance without the need of tedious autoregressive generation. There was no significant difference in the $KL\big(q(\mathbf{z}|x)||p(\mathbf{z})\big)$ between FAME and FAME No Concatenation. FAME use 10.85 nats in average to encode images, whereas FAME No Concatenation use 12.29 nats.

OMNIGLOT consists of 50 alphabets of handwritten characters, where each character has a limited amount of samples. Each character has high variance which makes it harder to fit a good generative model compared to MNIST. Table 3 presents the negative log-likelihood of FAME for OMNIGLOT and demonstrates significant improvement over previously published state-of-the-art. Figure 4b shows generated samples from the learned $\theta$ parameter space.

From Sønderby et al. (2016) we have seen that the LVAE is able to learn a much tighter $\mathcal{L}_1$ ELBO compared to the VAE. For the MNIST experiments, the $\mathcal{L}_1$ ELBO is at 80.11 nats

|  | NLL |
|---|---|
| IWAE (BURDA ET AL., 2015A) | 103.38 |
| LVAE (SØNDERBY ET AL., 2016) | 102.11 |
| RBM (BURDA ET AL., 2015B) | 100.46 |
| DVAE (ROLFE, 2017) | 97.43 |
| DRAW (GREGOR ET AL., 2015) | 96.50 |
| CONV DRAW (GREGOR ET AL., 2016) | 91.00 |
| VLAE CHEN ET AL. (2017) | 89.83 |
| FAME | **82.54** |

Figure 3: Negative log-likelihood performance on OMNIGLOT in nats. The evidence lower-bound is computed with 5000 importance weighted samples $\mathcal{L}_{5000}(\theta, \phi; x)$.

compared to the $\mathcal{L}_{5000}$ 77.82 nats. Similarly the OMNIGLOT $\mathcal{L}_1$ ELBO is 86.62 nats compared to 82.54 nats. This shows significant improvements when using importance weighted samples and indicates that the parameterization of the FAME can be done in a way so that the bound is even

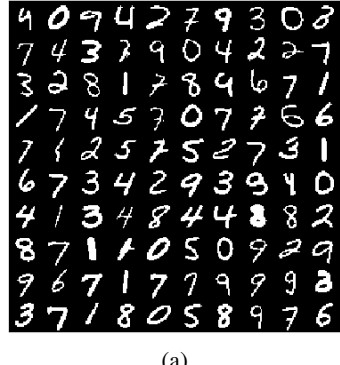 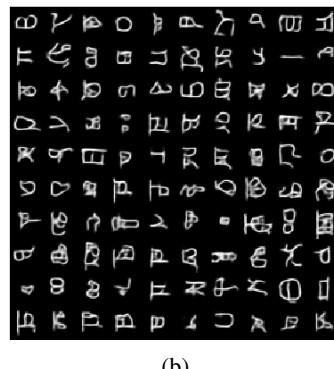

(a)           (b)

Figure 4: Random samples drawn from a $\mathcal{N}(0, I)$ distribution and propagated through the generative model of FAME for the dynamically binarized MNIST (a) and OMNIGLOT (b) dataset.

tighter. We also find that the top-most latent stochastic layer is not *collapsing* into its prior, since the $KL\big(q(z_5|x)||p(z_5)\big)$ is 5.04 nats for MNIST and 3.67 nats for OMNIGLOT.

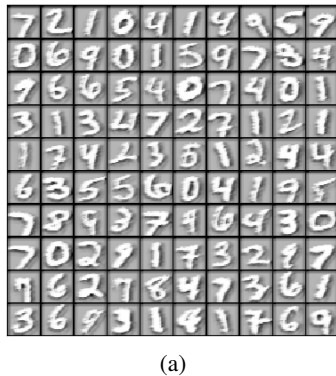 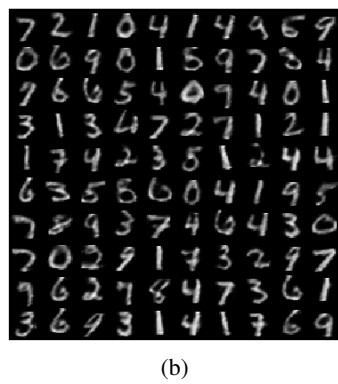

(a)           (b)

Figure 5: MNIST reconstructions when masking the output from the FAME stochastic variables (a) and the concatenated input image (b) prior to feeding them to the autoregressive PixelCNN. It is interesting to see how the edge information comes from the autoregressive dependency on the input image.

In order to analyze the contribution from the autoregressive decoder we experimented on masking the contribution from either the concatenated image or the output of the FAME decoder before feeding it into the PixelCNN layers (cf. Figure 1). In Figure 5a we see the results of reconstructing MNIST images when masking out the contribution from the stochastic variables and in Figure 5b we mask out the contribution from the concatenated input image.

### 3.2 GENERATIVE PERFORMANCE ON NATURAL IMAGES

We investigate the performance of FAME on two natural image datasets: CIFAR10 and ImageNet. Learning a generative model on natural images is more challenging, which is also why there are many *tricks* that can be done in regards to the autoregressive decoding (van den Oord et al., 2016a; Salimans et al., 2017; Chen et al., 2017). However, since we are interested in the additional expressiveness of a LVAE parameterization with convolutional stochastic latent variables, we have chosen a suboptimal architecture for the autoregressive decoding (cf. Table 1) (van den Oord et al., 2016b). An obvious improvement to the decoder would be to incorporate the PixelCNN++ (Salimans et al., 2017), but by using the simpler architecture we ensure that the improvements in log-likelihood is not a result of a strong autoregressive model.

From Table 3 we see the performance from FAME and FAME No Concatenation on the CIFAR10 dataset. Similarly to the gray-scaled images, FAME outperforms current state-of-the-art results sig-

| | BITS/DIM |
|---|---|
| UNIFORM DISTRIBUTION (VAN DEN OORD ET AL., 2016B) | 8.00 |
| DEEP DIFFUSION (SOHL-DICKSTEIN ET AL., 2015) | 5.40 |
| MULTIVARIATE GAUSSIAN (VAN DEN OORD ET AL., 2016B) | 4.70 |
| NICE (DINH ET AL., 2014) | 4.48 |
| DEEP GMMS (VAN DEN OORD & SCHRAUWEN, 2014) | 4.00 |
| CONV DRAW (GREGOR ET AL., 2016) | 3.58 |
| REAL NVP (DINH ET AL., 2016) | 3.49 |
| PIXELCNN (VAN DEN OORD ET AL., 2016B) | 3.14 |
| IAF VAE KINGMA ET AL. (2016) | 3.11 |
| GATED PIXELCNN (VAN DEN OORD ET AL., 2016A) | 3.03 |
| PIXELRNN (VAN DEN OORD ET AL., 2016B) | 3.00 |
| VLAE (CHEN ET AL., 2017) | 2.95 |
| PIXELCNN++ (SALIMANS ET AL., 2017) | 2.92 |
| FAME NO CONCATENATION | 2.98 |
| FAME | **2.75** |

Table 3: Negative $\log$-likelihood performance on CIFAR10 in bits/dim. The evidence lower-bound is computed with 1000 importance weighted samples $\mathcal{L}_{1000}(\theta, \phi; x)$.

| | BITS/DIM |
|---|---|
| CONV DRAW 32x32 (GREGOR ET AL., 2016) | 4.40 |
| CONV DRAW 64x64 (GREGOR ET AL., 2016) | 4.10 |
| REAL NVP 32x32 (DINH ET AL., 2016) | 4.28 |
| REAL NVP 64x64 (DINH ET AL., 2016) | 4.01 |
| PIXELVAE 64x64 (GULRAJANI ET AL., 2016) | 3.66 |
| PIXELRNN 32x32 (VAN DEN OORD ET AL., 2016B) | 3.86 |
| PIXELRNN 64x64 (VAN DEN OORD ET AL., 2016B) | 3.63 |
| GATED PIXELCNN 32x32 (VAN DEN OORD ET AL., 2016A) | 3.83 |
| GATED PIXELCNN 64x64 (VAN DEN OORD ET AL., 2016A) | 3.57 |
| FAME 32x32 | **3.23** |

Table 4: Negative $\log$-likelihood performance on ImageNet in bits/dim. The evidence lower-bound is computed with 1000 importance weighted samples $\mathcal{L}_{1000}(\theta, \phi; x)$.

nificantly. It is also interesting to see how FAME No Concatenation performs close to the previously published state-of-the-art results. Especially in the image space, this could prove interesting, since the FAME No Concatenation has no additional autoregressive runtime complexity. We only investigated the 32x32 ImageNet dataset, since the training time is significant and it outperformed the 64x64 models (cf. Table 4), whereas the previously published 64x64 ImageNet models consistently outperform their 32x32 counterpart. In Figure 2 we show samples from FAME on the CIFAR10 dataset. Similarly to previously published results it is difficult to analyze the performance from the samples. However, we can conclude that FAME is able to capture spatial correlations in the images for generating sharp samples. It is also interesting to see how it captures the contours of objects in the images.

## 4 CONCLUSION

We have presented FAME, an extension to the VAE that significantly improve state-of-the-art performance on standard benchmark datasets. By introducing feature map representations in the latent stochastic variables in addition to top-down inference we have shown that the model is able to capture representations of complex image distributions while utilizing a powerful autoregressive architecture as a decoder.

In order to analyze the contribution from the VAE as opposed to the autoregressive model, we have presented results without concatenating the input image when reconstructing and generating. This parameterization shows on par results with the previously state-of-the-art results without depending on the time consuming autoregressive generation.

Further directions for FAME is to (i) test it on larger image datasets with images of a higher resolution, (ii) expand the model to capture other data modalities such as audio and text, (iii) combine the model in a semi-supervised framework.

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
