# OpenReview forum: "Feature Map Variational Auto-Encoders"
_ICLR.cc/2018/Conference — Reject_

### Official Review · AnonReviewer1 · 2017-11-25
**Comments on the motivation, originality and experiments**

**Rating:** 5
**Confidence:** 4

**Review:**

The paper combines several recent advances on generative modelling including a ladder variational posterior and a PixelCNN decoder together with the proposed convolutional stochastic layers to boost the NLL results of the current VAEs. The numbers in the tables are good but I have several comments on the motivation, originality and experiments.

Most parts of the paper provide a detailed review of the literature. However, the resulting model is quite like a combination of the existing advances and the main contribution of the paper, i.e. the convolution stochastic layer, is not well discussed. Why should we introduce the convolution stochastic layers? Could the layers encode the spatial information better than a deterministic convolutional layer with the same architecture? What's the exact challenge of training VAEs addressed by the convolution stochastic layer? Please strengthen the motivation and originality of the paper.

Though the results are good, I still wonder what is the exact contribution of the convolutional stochastic layers to the NLL results?  Can the authors provide some results without the ladder variational posterior and the PixelCNN decoder on both the gray-scaled and the natural images?

According to the experimental setting in the Section 3 (Page 5 Paragraph 2), "In case of gray-scaled images the stochastic latent layers are dense with sizes 64, 32, 16, 8, 4 (equivalent to Sønderby et al. (2016)) and for the natural images they are spatial (cf. Table 1). There was no significant difference when using feature maps (as compared to dense layers) for modelling gray-scaled images." there is no stochastic convolutional layer.  Then is there anything new in FAME on the gray images? Furthermore, how could FAME advance the previous state-of-the-art? It seems because of other factors instead of the stochastic convolutional layer.

The results on the natural images are not complete. Please present the generation results on the ImageNet dataset and the reconstruction results on both the CIFAR10 and ImageNet datasets. The quality of the samples on the CIFAR10 dataset seems not competitive to the baseline papers listed in the table. Though the visual quality does not necessarily agree with the NLL results but such large gap is still strange. Besides, why FAME can obtain both good NLL and generation results on the MNIST and OMNIGLOT datasets when there is no stochastic convolutional layer? Meanwhile, why FAME cannot obtain good generation results on the CIFAR10 dataset? Is it because there is a lot randomness in the stochastic convolutional layer? It is better to provide further analysis and it is not safe to say that the stochastic convolutional layer helps learn better latent representations based on only the NNL results.

Minor things:

Please rewrite the sentence "When performing reconstructions during training ... while also using the stochastic latent variables z = z 1 , ..., z L." in the caption of Figure 1.

---

### Official Review · AnonReviewer3 · 2017-11-27
**limited novelty. possibly incorrect?**

**Rating:** 3
**Confidence:** 3

**Review:**

The description of the proposed method is very unclear. From the paper it is very difficult to make out exactly what architecture is proposed. I understand that the prior on the z_i in each layer is a pixel-cnn, but what is the posterior? Equations 8 and 9 would suggest it is of the same form (pixel-cnn) but this would be much too slow to sample during training. I'm guessing it is just a factorized Gaussian, with a separate factorized Gaussian pseudo-prior? That is, in figure 1 all solid lines are factorized Gaussians and all dashed lines are pixel-cnns?

* The word "layers" is sometimes used to refer to latent variables z, and sometimes to parameterized neural network layers in the encoder and decoder. E.g. "The top stochastic layer z_L in FAME is a fully-connected dense layer". No, z_L is a vector of latent variables. Are you saying the encoder produces it using a fully-connected layer?
* Section 2.2 starts talking about "deterministic layers h". Are these part of the encoder or decoder? What is meant by "number of layers connecting the stochastic latent variables"?
* Section 2.3: What is meant by "reconstruction data"?

If my understanding of the method is correct, the novelty is limited. Autoregressive priors were used previously in e.g. the Lossy VAE by Chen et al. and IAF-VAE by Kingma et al. The reported likelihood results are very impressive though, and would be reason for acceptance if correct. However, the quality of the sampled images shown for CIFAR-10 doesn't match the reported likelihood. There are multiple possible reasons for this, but after skimming the code I believe it might be due to a faulty implementation of the variational lower bound. Instead of calculating all quantities in the log domain, the code takes explicit logs and exponents and stabilizes them by adding small quantities "eps": this is not guaranteed to give the right result. Please fix this and re-run your experiments. (I.e. in _loss.py don't use x/(exp(y)+eps) but instead use x*exp(-y). Don't use log(var+eps) with var=softplus(x), but instead use var=softplus(x)+eps or parameterize the variance directly in the log domain).

---

> ### Public Comment · ~Yoon_Kim1 · 2017-12-07
> **.**
>
> [EDIT]: I just realized I had a really silly bug in my implementation. Please disregard my previous posts regarding successful replication.
>
> Sorry for adding noise to the process!

---

> > ### Public Comment · (anonymous) · 2017-12-15
> > **The nature of the bug**
> >
> > Thank you for the update. Could you say what the bug was, even if it was silly? This would allow other researchers (including the authors of the paper) to make sure that they don't have this bug in their code as well.

---

> > > ### Public Comment · ~Yoon_Kim1 · 2017-12-15
> > > **.**
> > >
> > > Sure, it's incredibly silly/embarrassing: I didn't realize that unlike MNIST/OMNIGLOT, CIFAR numbers were in bits and not nats!

---

### Official Review · AnonReviewer2 · 2017-11-27
**Review: Impressive experimental results but low novelty**

**Rating:** 6
**Confidence:** 4

**Review:**

Update:  In light of Yoon Kim's retraction of replication, I've downgraded my score until the authors provide further validation (i.e. CIFAR and ImageNet samples).

Summary

This paper proposes VAE modifications that allow for the use multiple layers of latent variables.  The modifications are: (1) a shared en/decoder parametrization as used in the Ladder VAE [1], (2) the latent variable parameters are functions of a CNN, and (3) use of a PixelCNN decoder [2] that is fed both the last layer of stochastic variables and the input image, as done in [3].  Negative log likelihood (NLL) results on CIFAR 10, binarized MNIST (dynamic and static), OMNIGLOT, and ImageNet (32x32) are reported.  Samples are shown for CIFAR 10, MNIST, and OMNIGLOT.


Evaluation

Pros:  The paper’s primary contribution is experimental: SOTA results are achieved for nearly every benchmark image dataset (the exception being statically binarized MNIST, which is only .28 nats off).  This experimental feat is quite impressive, and moreover, in the comments on OpenReview, Yoon Kim claims to have replicated the CIFAR result.  I commend the authors for making their code available already via DropBox.  Lastly, I like how the authors isolated the effect of the concatenation via the ‘FAME No Concatenation’ results.

Cons:  The paper provides little novelty in terms of model or algorithmic design, as using a CNN to parametrize the latent variables is the only model detail unique to this paper.  In terms of experiments, the CIFAR samples look a bit blurry for the reported NLL (as others have mentioned in the OpenReview comments).  I find the authors’ claim that FAME is performing superior global modeling interesting.  Is there a way to support this experimentally?  Also, I would have liked to see results w/o the CNN parametrization; how important was this choice?


Conclusion

While the paper's conceptual novelty is low, the engineering and experimental work required (to combine the three ideas discussed in the summary and evaluate the model on every benchmark image dataset) is commendable.  I recommend the paper’s acceptance for this reason.


[1]  C. Sonderby et al., “Ladder Variational Autoencoders.”  NIPS 2016.
[2]  A. van den Oord et al., “Conditional Image Generation with PixelCNN Decoders.” ArXiv 2016.
[3]  I. Gulrajani et al., “PixelVAE: A Latent Variable Model for Natural Images.”  ICLR 2017.

---

### Public Comment · (anonymous) · 2017-11-01
**Questions about your results**

Intrigued by the claim of state-of-the-art results, I read the paper and noticed several things about the results that don't look right.

The Omniglot samples in Figure 4 don't look binary. Are you showing the probabilities instead of the actual samples? Which version of the Omniglot dataset did you use and how did you preprocess it?

The MNIST samples in Figure 4 do look binary, but their edges are far too smooth for stochastically binarized MNIST. In other words, they don't actually look like the data: just compare them to samples on Figure 1 in the VLAE paper (https://arxiv.org/abs/1611.02731). Did you sample each pixel or did you just use the most probable value?

The CIFAR10 samples in Figure 2 have very little local detail and are not nearly sharp and structured enough to correspond to the 2.75 bits/dim result reported in the paper. In my experience, a model generating samples like this should get 3.4 bits/dim at best. What kind of test NLL estimates do you get with a single sample on CIFAR10 and ImageNet?

Finally, given that you seem to have the best 32x32 ImageNet result by a huge margin it seems odd not to include any samples from this model. Why did you omit them? While sample quality is just one aspect of model performance, in my experience a large discrepancy between sample quality and NLL in a VAE-like model usually means that there's a bug in the code.

---

> ### Public Comment · ~Yoon_Kim1 · 2017-11-01
> **.**
>
> I too found the CIFAR results remarkable, so I replicated it, and I was able to match the ~2.8 number (with slightly different architecture/hyperparameters than was used in the paper).
>
> Really nice work!

---

> > ### Public Comment · (anonymous) · 2017-11-02
> > **Computing the test NLL**
> >
> > Thanks a lot for sharing your replication experience. Do your CIFAR10 samples look substantially different from the ones in the paper? When computing the test NLL estimate, do you use the batchnorm statistics from the training set or the test set?

---

> > > ### Public Comment · ~Yoon_Kim1 · 2017-11-02
> > > **.**
> > >
> > > No problem! I haven't checked the samples and I use the batchnorm statistics from the training set only.

---

> > > > ### Author Response · Authors · 2017-11-14
> > > > **Answering the questions to the results**
> > > >
> > > > Dear Yoon Kim and Anonymous,
> > > >
> > > > Thank you for your interest in the paper. We have taken the comments seriously and thoroughly reviewed the code without finding any bugs. We have retrained the models and are confident in the results given in the Tables. We actually also found slight improvements compared to the results reported.
> > > >
> > > > First of all we would like to answer the questions formulated on the 1st of November by anonymous:
> > > >
> > > > Q1. The Omniglot samples in Figure 4 don't look binary. Are you showing the probabilities instead of the actual samples? Which version of the Omniglot dataset did you use and how did you preprocess it?
> > > >
> > > > Yes, we are showing the probabilities. We will include the stochastically binarized samples in the revised version. Unfortunately it is not possible to include them in this OpenReview format, so please see: https://imgur.com/gallery/NRvxO. From the plots we can see comparable results to VLAE. It is hard to distinguish whether one is better over the other by only evaluating the samples, hence the log-likelihood results in the tables should tell the full story.
> > > >
> > > > Q2. The MNIST samples in Figure 4 do look binary, but their edges are far too smooth for stochastically binarized MNIST. In other words, they don't actually look like the data: just compare them to samples on Figure 1 in the VLAE paper (https://arxiv.org/abs/1611.02731). Did you sample each pixel or did you just use the most probable value?
> > > >
> > > > You are right, we used the most probable value. We will include the stochastically binarized samples in the revised version. Find them here: https://imgur.com/gallery/NRvxO .
> > > >
> > > > Q3. The CIFAR10 samples in Figure 2 have very little local detail and are not nearly sharp and structured enough to correspond to the 2.75 bits/dim result reported in the paper. In my experience, a model generating samples like this should get 3.4 bits/dim at best. What kind of test NLL estimates do you get with a single sample on CIFAR10 and ImageNet?
> > > >
> > > > We do agree that these samples do not have as much local detail as the samples in VLAE and PixelCNN++. However, we have a very simple PixelCNN parameterization without the R->G->B dependency and all of the additional contributions as the PixelCNN++/VLAE papers have. In our approach the local structure is worse but the global modeling is better. We interpret the better test likelihood score as a sign that visual appeal is not the ultimate way to judge the model. We expect that including a better autoregressive model in FAME will give us the best of both worlds.
> > > >
> > > > For the camera-ready version we will train FAME with a more complex autoregressive model and visualize the generated images in the final paper for both ImageNet and CIFAR10. We didn't do this experiment, since we did not have the time before the deadline and we were more interested in answering the question to why the additional VAE parameterization in PixelVAE and VLAE didn't give a better bound.
> > > >
> > > > Last but not least we will publish the code on Github upon publishing the paper.

---

> > > > > ### Comment · AnonReviewer3 · 2017-11-22
> > > > > **possible to share code anonymously?**
> > > > >
> > > > > Would it be possible to somehow share the code before the review period ends? Currently I also have a very hard time believing the reported 2.75 bits per dim number on CIFAR-10.

---

> > > > > > ### Author Response · Authors · 2017-11-23
> > > > > > **General comments and code sharing**
> > > > > >
> > > > > > Dear anonymous and AnonReviewer3,
> > > > > >
> > > > > > * Thank your for spotting the sub-quality sample. We have identified a possible error. We had forgotten to sample the softmax from the auto-regresssive part of the model. This of course have a negative influence on the sample quality but does not affect the test log-likelihood calculation. We will provide a follow-up on this a bit later with new samples.
> > > > > >
> > > > > > * We provide the complete code here (Python 3 & Tensorflow 1.2):  https://www.dropbox.com/s/wjhhxff0b0np6xi/FAME-implementation.zip?dl=0. We will also provide the code in a Github repo later.
> > > > > >
> > > > > > * We have trained a PixelCNN with an equivalent architecture as the one used for FAME (no R->G->B dependency) and achieved a NLL at 3.34 bits/dim. So we are confident that our code is working properly.
> > > > > >
> > > > > > * Finally we would like to note that we forgot to add a “log” in the equation following Eq. 6 in the paper.

---

> > > > > > > ### Comment · AnonReviewer3 · 2017-11-23
> > > > > > > **spotted a bug**
> > > > > > >
> > > > > > > Thanks for sharing the code! I went through some of the files super quickly, and I seem to spot at least 1 bug: In StochasticGaussian() you create a random distribution z ~ N(m,s), and then you produce output z' = m + s*z. Seems like you're transforming the standard normal variable twice? Let me know if I'm wrong.

---

> > > > > > > > ### Author Response · Authors · 2017-11-23
> > > > > > > > **Not a bug**
> > > > > > > >
> > > > > > > > Dear AnonReviewer3,
> > > > > > > >
> > > > > > > > First we generate a random tensor N(0,I)->'eps' (line 47) that has the same shape as the input then we calculate z (line 51) by applying the reparameterization trick: https://arxiv.org/pdf/1312.6114.pdf.

---

> > > > > > > > > ### Comment · AnonReviewer3 · 2017-11-23
> > > > > > > > > **line 47**
> > > > > > > > >
> > > > > > > > > this is line 47: eps = tf.random_normal(tf.shape(input_mean),mean=self.mean, stddev=np.sqrt(self.var), seed=self.seed, name=self.name)
> > > > > > > > >
> > > > > > > > > Are you saying self.mean=0 and self.var=1 always?

---

> > > > > > > > > > ### Author Response · Authors · 2017-11-23
> > > > > > > > > > **line 47**
> > > > > > > > > >
> > > > > > > > > > Yes. In _fame.py lines: 60, 99 and 105, you can see that I call the function using the default arguments mean=0. and var=1.
> > > > > > > > > >
> > > > > > > > > > Please note that there is a difference between self.mean, self.var and input_mean, input_var.

---

> > > > > > > > > > > ### Comment · AnonReviewer3 · 2017-11-23
> > > > > > > > > > > **another thing to try**
> > > > > > > > > > >
> > > > > > > > > > > ok, perhaps I was too quick there.
> > > > > > > > > > >
> > > > > > > > > > > Could you try evaluating a trained model with eps=0 in the variational bound instead of eps=1e-8? Since both the prior and posterior are learned, the model might learn to take advantage of your stability measures (this is not the best way of implementing this). If this matters (i.e. if the eps actually does something) the bound would be bad.

---

> > > > > ### Public Comment · (anonymous) · 2017-11-22
> > > > > **CIFAR10 results**
> > > > >
> > > > > Thank you for your detailed response. The MNIST and Omniglot samples you provided look reasonable and appear consistent with the scores you report on the datasets.
> > > > >
> > > > > I'm still puzzled by the apparent discrepancy between the sample quality and the test log-likelihood estimates on CIFAR10. To me it looks like the top-left pixel in all the samples in Figure 2 is white, which suggests that something is wrong with either training or sampling. You might want to check whether your PixelCNN implementation is correct for RGB data, e.g. conditioning is consistent between training and sampling. What do the samples look like if you remove the latent variables and train just the PixelCNN component?

---

> > > ### Public Comment · ~Yoon_Kim1 · 2017-11-06
> > > **.**
> > >
> > > Interestingly, I do find that the samples are quite a bit blurrier than other papers that achieve higher bits/dim (e.g. VLAE/PixelCNN++, etc.).
> > >
> > > Authors: As the above poster suggested, I would be curious to see the ImageNet samples as well.
> > > Also, how does the KL look for CIFAR10? What about reconstructions?
> > >
> > > It could also mean that our intuition regarding bits/dim translating to higher quality samples is not necessarily true, e.g. due to teacher-forced training vs sampling-based generation. Or it could simply be a bug on my part...
> > >
> > > Incidentally, I was able to get the bits/dim down ~2.7 by playing around with the hyperparameters a bit more. For me these seemed to help:
> > >
> > > - learning the prior (log) variances
> > > - using a higher dimensional latent dimension at each stage (I use 32 at each stage, and my first latent map is at the 8 x 8 resolution)

---

### Public Comment · (anonymous) · 2017-12-15
**Correctness and samples**

In light of the independent replication claim being retracted below, could the authors comment whether they still believe that the results reported in the paper are correct? If so, could you post some CIFAR10 and ImageNet samples obtained after fixing the sampling bug mentioned below?

---

### Author Response · Authors · 2018-01-05
**Authors final comments**

Dear reviewers,

Thank you for all of your useful feedback. We have used this rebuttal period to investigate our results and have found that

1) the grayscale MNIST and OMNIGLOT result hold and
2) we too had a bug in the AR model part for the natural color images.

We have corrected the bug by now and the samples look much better. However, we won’t be able to update the results in due time, which is why we completely understand you not accepting the paper in its current format. We plan to submit to ICML and apologize for the inconvenience.

---

### Decision · Program_Chairs · 2018-01-29
**ICLR 2018 Conference Acceptance Decision**

**Decision:**

Reject

**Comment:**

The paper proposes a VAE variant by embedding spatial information with multiple layers of latent variables. Although the paper reports state-of-the-art results on multiple datasets, some results may be due to a bug. This has been discussed, and the author acknowledges the bug. We hope the problem can be fixed, and the paper reconsidered at another venue.